# LLM-Supported Safety Annotation in High-Risk Environments

Mohammad Eskandari*, Murali Indukuri*, Stephanie Lukin†, Cynthia Matuszek*
*University of Maryland, Baltimore County, Baltimore, MD, USA
{eskandari, muralii1, cmat}@umbc.edu
†Army Research Lab, Adelphi, MD, USA
stephanie.m.lukin.civ@army.mil

*Abstract*—This paper explores how large language model-based robots assist in detecting anomalies in high-risk environments and how users perceive their usability and reliability in a safe virtual environment. We present a system where a robot using a state-of-the-art vision-language model autonomously annotates potential hazards in a virtual world. The system provides users with contextual safety information via a VR interface. We conducted a user study to evaluate the system's performance across metrics such as trust, user satisfaction, and efficiency. Results demonstrated high user satisfaction and clear hazard communication, while trust remained moderate.

*Index Terms*—Large Language Models, Virtual Reality, Human-Robot Interaction, Hazard Detection, Safety Annotation

## I. INTRODUCTION

Human-Robot Interaction (HRI) is becoming increasingly important in high-risk environments, from search and rescue operations to industrial safety and space exploration [1], [2]. These domains often involve safety standards that require individuals to follow strict regulations, such as those set by the Occupational Safety and Health Administration (OSHA). Traditional hazard mitigation approaches can be inefficient in ensuring such regulations [3]. For example, human fatigue can increase the risk of turning minor hazards into serious incidents, and information overload may reduce human awareness of useful data during disasters.

In this paper, we describe a tool intended to reduce these risks by improving hazard awareness and risk management [4]. We imagine this tool to be used as part of a scenario in which an autonomous robot such as a drone is released to build up a virtual map of a high risk environment. Our system can then autonomously label hazards in that environment, allowing a responder to explore the annotated environment in virtual reality, allowing for safe decision making and situational awareness. Figure 1 shows an example of an annotated environment. Broadly speaking, our approach works by taking images of areas in a virtual environment and presenting them to a large vision-and-language model (VLM), with instructions to identify any risks or hazards present. The environment is then annotated with the responses in the form of pop-ups that appear when a person exploring the space approaches a hazard. We refer to these annotations as points of interest (POIs).

This work was supported by the National Science Foundation under Award #001583-00001 (CAREER: Robots and Language in Human Spaces).

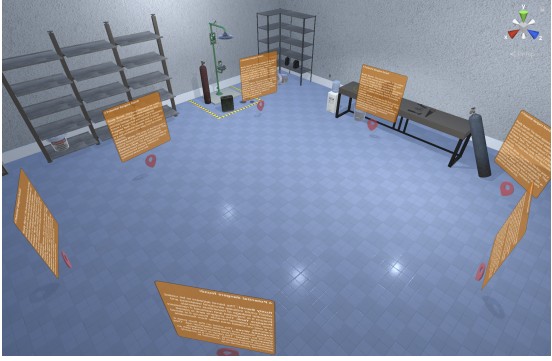

(a) A high level view of an annotated environment.

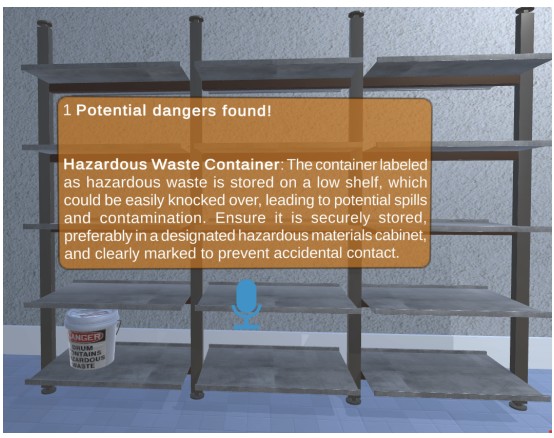

(b) A participant's point-of-view showing a hazard.

Fig. 1: An example of a hazardous environment used in our experiments (a maker-space after an earthquake), with possible hazards autonomously annotated by interaction with a large vision-and-language model.

Risk management is typically broken down into mitigation, preparedness, response, and recovery [4]. Our focus is on the proactive side of risk management, mitigation and preparedness, as preventing people from encountering hazards unexpectedly is ultimately more effective than reacting to them. At the same time, the sometimes overwhelming flood of information during disaster response [4] means that even reactive strategies may benefit from structured hazard annotation. Our hypothesis is that this interface will increase situational

awareness for people who must respond to potentially dangerous situations, allowing information gathered by a robot to be explored and presented in a safe, intuitive way.

Despite growing interest in robotics for emergency response, industrial safety, and military operations, human-robot teams still struggle with adaptability, coordination, and clear communication [5], [6]. One approach to managing human-robot interfaces is using natural language to enable robots to comprehend complex, context-rich commands so that non-expert users have an intuitive way of accessing information and interacting with a robot teammate [7]. Recent advances in Large Language Models (LLMs) have been widely used to help robots understand [8] and plan over [9] human-like language. Similarly, large Vision-and-Language Models (VLMs) are particularly capable in visual reasoning tasks, and are currently being studied for purposes such as safety planning, navigation, and scene understanding [10].

There are a number of high-level ways in which LLMs and VLMs can be tied into HRI systems. They can be "Scarecrows [11]," stand-ins for more principled approaches that will ultimately be replaced by more carefully developed technologies, or they can be a carefully selected component of a larger system in their own right. Our use of VLMs is in the latter category; we use careful prompt design and limited natural language interactions to construct a system that makes use of the power of large pre-trained models without the problems associated with direct human/model communication. Unlike studies that only discuss the interaction between humans and AI-enabled systems [11]–[13], we design and test this interface for a possible setting, and incorporate user feedback to refine it for practical future use.

Virtual and augmented reality are progressively more integral to HRI research, and well-designed interfaces for them improve interactions between intelligent systems and operators [12]. We use the Robot Interactions in Virtual Reality (RIVR) simulator [14], a simulation environment designed to support Human-Robot Interaction in VR. RIVR's combination of VR interaction support, ROS robot management, and Unity environments allows us to implement an immersive user-friendly system of annotations for an intuitive user experience.

**Hypotheses:** Building on these foundations, we hypothesize that LLM-supported robots can identify at least 80% of hazardous items in a given space (H1). Furthermore, we expect that people will generally feel improved situational awareness when navigating a scene with annotations as opposed to an un-annotated scene (H2).

To investigate these hypotheses, we present an LLM-powered system that autonomously annotates hazards in VR, aiming to improve robotic hazard communication, reduce operator cognitive overload, and enhance proactive risk management in HRI. We evaluate this system in a user study to assess its effectiveness in improving hazard perception and user comfort in VR-based HRI scenarios.

Specifically, our contributions in this study are as follows:

1) We present a system designed to autonomously annotate hazards in Virtual Reality environments using VLMs that provides an open-source framework for robotic hazard annotation, including structured prompts, messages, and Python scripts, integrated with robotic platforms via ROS and the RIVR simulation.
2) We present and evaluate an immersive interface featuring virtual markers with auditory message playback, enhancing user interaction and accessibility within the framework.
3) We demonstrate that use of this framework provides improved user comfort and an increased sense of situational awareness for participants interacting with a virtual setting, in this case a maker-space after an earthquake.

## II. BACKGROUND AND RELATED WORK

### A. VLM-based Approaches

VLMs have demonstrated wisdom-of-the-crowd capabilities in 3D reasoning [15], [16] and show promise in specialized tasks such as hazard analysis and disaster response [17]–[20]. These models have so much information that to extract desired information from them, it is encouraged to use guidelines such as system definitions and "lines of inquiry," even if the system's output degrades as complexity increases [17]. Apart from the need for well-structured prompts, key challenges also include hallucination risks and computational costs, reducing which is particularly prioritized in critical scenarios.

For disaster response, large language models can effectively analyze details of a scenario based on pictures of the scene [18]–[20]. This includes abilities such as extracting water depth information from images of floods [18], [19]. A data pipeline can be constructed with existing tools and cloud services like Google's reverse geocoding, Microsoft Azure, various GIS systems, and an LLM at the head to generate detailed reports of disasters in near-real time to aid first-responders [19]. Using these data pipelines, VLMs can augment or replace the role of people in reporting emergencies, whose ability to do so may be impaired in disastrous situations [19]. This is further supported by evidence than VLM analysis of disasters closely matches that of humans [18]. There has also been significant research regarding VLMs' ability to understand and annotate 3D scenes, an essential component of hazard analysis. It has been shown that zero-shot prompting techniques can uncover latent geometric and spatial reasoning capabilities and improve VLMs' ability to answer questions about a scene [15], [16]. Furthermore, VLMs can be used to generate position data that can then be used by standard LLMs to perform similar 3D reasoning tasks [16]. Although VLMs perform better in scene understanding than their LLM counterparts, they tend to have a heavy bias towards a text prompt rather than the image prompt when both are included [22]. As will be discussed later, our work confirms this observation that VLM output quality depends highly on the detail of the text prompt.

### B. Non-VLM Approaches

As alternate approaches to analyzing hazards, researchers typically train models from scratch and use standard image

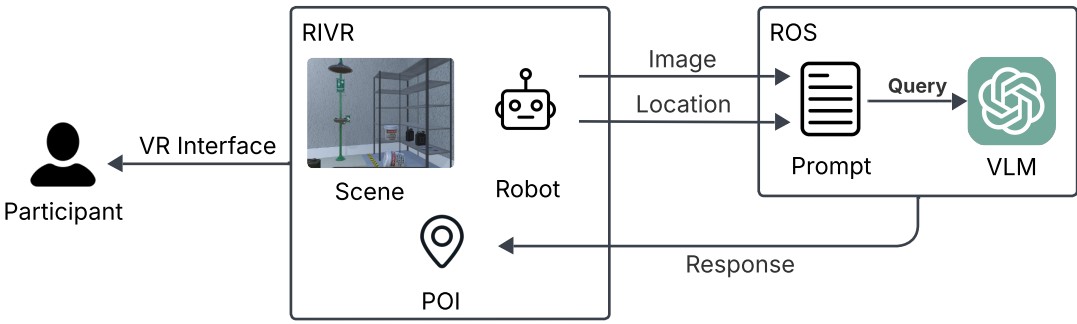

Fig. 2: An overview of our system's architecture. The RIVR simulator communicates with ROS, and a VLM server responds to the robot's queries about hazards and generating points of interest. (Diagram created in Lucid [21])

classifiers to drive such analysis [23], [24]. These approaches use deep learning methods to identify hazards, such that given RGB-D images of a scene, they generate labeled point clouds containing semantic hazard information like fire or tripping hazards [24]. Furthermore, standard classifiers like YOLO [25], [26] can detect the presence or absence of safety equipment in a scenario and unsafe behaviors like people entering a hazardous area. Extracted information from these object detection models can be used to determine hazards and take steps to avoid them [23], [27]. However, their ability to analyze risks is limited by their training data, as they are inherently closed-set models, making them less adaptable to scenarios outside their training scope. In this paper, we choose to employ a VLM in particular to address the domain adaption challenges where our VLM prompt and pipeline might be applied to new scenes.

### C. VR for Training and Hazard Analysis

VR can perform as an effective medium for hazard analysis and training in dangerous scenarios compared to conventional 2D methods like images on paper on presentations [28]–[31]. Simulators already exist for harsh environments like space, but often do not have a good interface for human interaction; VR can be used to make training in such environments more stimulating [28]. Generalizing to other industries like mining, electronics, and construction, VR can be used to increase the immersion and presence of trainees in managing hazards; it is more immersive than CAVE systems or desktop VR, making the medium an effective tool [29], [30]. Training presented in this manner has shown to be retained better over a period of several weeks [29]. Although this work does not directly address training, this existing work demonstrates the potential of VR as an effective interface for understanding risks.

In addition to training, VR has also proven effective in aiding hazard analysis. Simulations often require simplification of models at the risk of omitting information regarding hazards [31]. Increasing the fidelity of a simulation and including human models increase the number of risks identified by participants [31]. The ability for participants to perceive the true scale of objects may also aid in this process [30], [31].

However, it should be noted that the effectiveness of VR can be limited by the experience of the participants [30].

Preceding work has also examined VR as an interface for presenting data from robotic exploration of a hazardous scenario [32]. It was shown that auditory cues can be generated by robots to help participants recognize hazards in a simulated scenario. However, a large number of sounds occurring at the same time proved stressful. Our approach offloads the duty of hazard recognition to VLMs; we hope this is a more comfortable user experience.

We note that there is an absence of research studying how LLM and VLM-guided robots can help with hazard analysis and annotation of anomalies. Specifically, we study the ability of GPT-4o to evaluate anomalies given images of a hazardous scenario and propose a way to present this information to users in VR.

### III. APPROACH

As shown in Figure 2, we design a system where the RIVR simulator is integrated with ROS and the VLM annotation server. The annotations are generated using GPT-4o [33], which is widely utilized in research due to its ease of use, broad API capabilities, and multimodal processing. The VR application is developed in Unity (version 2019.4.21f1), utilizing the High-Definition Rendering Pipeline (HDRP) for high-quality graphics.

### A. System Design

Various high-risk environments were considered, including a train derailment, a maker-space after an earthquake, and a construction site equipment failure. The makerspace setting was chosen for our experiments because the indoor setting facilitates robot exploration of the environment in order to build the virtual world. It also offers a diverse range of potential hazards (hazardous chemicals, pressurized gas, tools, obstructed eyewash and fire suppressant stations). We simulated robot exploration of the environment with a Husky UGV with a full suite of sensors. The availability and efficiency of the Husky makes the sim2real process easier. Autonomous exploration through SLAM is yet to be fully integrated into our

system; at this time, the robot exploration of the environment to be annotated is manual.

During the robot's exploration of the room, it captures photos at a fixed rate and makes queries to the VLM service about safety issues in the image with an engineered prompt. After the robot has made a complete circuit of the environment and sent a large number of images to the VLM server to identify points of interest, any identified hazards are inserted as Points of Interest into the environment. We define POIs as virtual markers that draw users' attention to certain areas in the simulator. These markers have two states: idle and activated. When idle, their visibility is a transparent red pin bouncing up and down; once a person enters the proximity of a marker, it is activated, which brings up an information panel designed to present information in an intuitive way to help decision making. Pilot studies on the system suggested that some users found it difficult to read in virtual reality, motivating us to add a virtual button on the panel that allows users to listen to the annotations being read out loud. To do this, we used the audio file provided by ChatGPT-4o, generated separately for each response.

### B. Prompting VLMs

Prompting is providing the VLM model with a natural language instruction that explains how to complete a task. It stands out as a promising, lightweight solution for eliciting the capabilities of LLMs and VLMs without model parameter tuning [34]. To identify the most effective prompt for hazard identification, we experimented with a range of queries—from simple, short prompts (e.g., 'Identify hazards in this image'), to more complex, detailed instructions that incorporated role definitions, context, and safety standards.

To evaluate prompt effectiveness, we conducted a small pilot study with ten participants who rated the safety reports generated by a Vision-Language Model (VLM) based on different prompts. Participants assessed the reports on several criteria, including hallucination in hazard detection, overall acceptability, precision in hazard detection, and the influence of different fields of view (FOVs) on report quality. Our initial prompts were only a few dozen tokens and resembled human commands; for instance, one of them was "This is a photo of a maker-space. Do you see any misplaced, potentially dangerous, or hazardous items?", which yielded ambiguous outputs and lower accuracy in hazard detection. By gradually increasing the prompt's complexity—specifying the system's role as a safety inspector, outlining the available robot actions, and structuring the output—we observed a marked improvement in the precision and clarity of the model's responses.

To keep our system generalizable to other environments with different safety concerns, we avoid specific details and scenario-dependent variables. Thus, we keep it simple by stating the room is a high-risk environment with potential hazardous items and situations. The main linguistic components of the prompt for the VLM, as used for the user study, are shown below; these prompts are paired with an image that the system can analyze for hazards:

---

**Prompt: Robot Actions**

The set of robot actions you can choose from is: {Generate POI, Do Not Generate POI}.

---

**Prompt: System Role**

You are an expert safety inspector specializing in hazard identification. Your primary role is to accurately and thoroughly identify potentially hazardous items in various environments. Provide clear, detailed explanations for why each item might be hazardous and suggest appropriate safety measures, ensuring high standards of safety are maintained. Respond concisely.

---

**Prompt: Output Structure**

Respond in the form of a JSON object, based on the class provided below.
**class HazardReport:**
int hazards_Found;
string safetyMessage;
Vector3 coordinates;
string image;
bool generatePOI;
Only a single JSON object with no extra words. Follow the given instructions to populate it.

1) Populate `hazards_found` with the count of hazards found.
2) Populate `safetyMessage` with hazard descriptions, starting with each hazard's name, followed by a caution note explaining why it is dangerous and what safety measures should be taken. Begin each hazard on a new line using \n to create the line break.
3) Populate `coordinates` with the location provided in {`data.position`}.
4) Populate `image` with the path {`data.image`}.
5) If at least one hazard is identified, set `generatePOI` to `true`, otherwise `false`.

---

**Response Example**

```
{
    "hazards_found": 1,
    "safetyMessage":
    "1 Potential dangers found!
    Hazardous Waste Container: The
    container labeled as hazardous
    waste is stored on a low shelf...",
    "coordinates": {12.3, 56.7, 0.0},
    "image": "image.png",
    "generatePOI": true
}
```

## IV. User Study

HRI is a field in which robotic systems are designed and evaluated for use by or with humans [35]. To assess this system, an IRB-approved user study was conducted. The study used survey-based evaluations to determine the feasibility and usefulness of the interface for people of various technical backgrounds. In this user study, the baseline condition is an unannotated scene compared against the LLM-powered annotated scene. After listening to the details of the experiment, they wear the VR headset (HTC VIVE Pro 2) to step into the virtual environment. In the simulation, first a tutorial scene is displayed in which participants can familiarize themselves with the VR technology, teleportation, and interaction with example POIs. After participants complete the tutorial, the system displays the fully annotated makerspace.

We recruit 12 participants from a university setting, representing a diverse group in terms of age, gender, and prior experience with VR. The participants included 7 females and 5 males, and ages ranged from 22 to 50 years, with the majority (8 participants) aged between 22 and 32 years. Four participants were aged between 25 and 50 years. Among the participants, 9 had no prior experience with VR, while 3 were familiar with the technology. We believe this demographic sample is representative of gender, age, and VR experience, allowing for a more comprehensive understanding of the system's impact across different user backgrounds.

Participants rated their experience on a **5-point Likert scale**, across the following key dimensions:

---

### Evaluation Questions

1) **Clarity** – How clearly were hazards marked in the virtual environment?
2) **Text Effectiveness** – How well did the system convey safety information through annotations?
3) **Trust in System** – How confident are you in the system's ability to accurately detect hazards?
4) **UI Usefulness** – How useful is the VR interface for improving safety awareness?
5) **Future Use** – How likely are you to use this system in real-world scenarios?

---

All work was performed after the lead authors' Institutional Review Board (IRB) evaluated the study protocols and all materials. Participants were asked to sign a consent form before beginning the familiarization task, and were informed clearly that they could withdraw from the task at any time without penalty.

## V. Results

The average ratings for each of the five key aspects were consistently high, reflecting the system's effective performance, clear annotations, and strong user engagement. Clarity received the highest mean rating (4.6, SD = 0.65), indicating that most participants found the hazard indicators clear. Text Effectiveness was rated 4.25 (SD = 0.62), with suggestions for

improved positioning and clarity. Trust in the System scored 4.16 (SD = 0.83), reflecting moderate trust, with real-world application identified as critical for confidence building. UI Usefulness (4.58, SD = 0.79) and Future Use (4.5, SD = 0.79) highlight the system's positive impression and strong potential for operational adoption (See fig. 3).

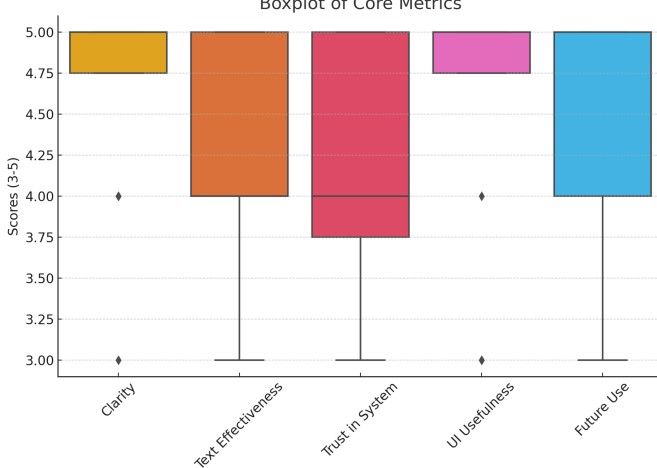

Fig. 3: Boxplot of core metrics evaluated in the user study, with mean scores ranging from 4.16 to 4.6 on a 5-point scale.

Additionally, participants were asked to select whether they **preferred the annotated or unannotated environment** for safety assessment. Only one participant preferred the unannotated scene over the annotated one. Finally, an **open-ended response section** allowed them to provide **comments and suggestions** for system improvements. This helped us to better understand our system's shortcomings. Most people showed much enthusiasm towards our system, working with robots, and VR. Some suggested that the information panel's height should be user-adjustable for easier reading. Additionally, they recommended adding a minimize feature to allow users to hide or show the panel as needed. A few recommended integration of other sensors like temperature, pressure, and sound. One suggested to use more recent VR devices such as Quest 3 for a wireless experience. The collected responses were analyzed to identify trends, with correlation analysis used to determine relationships between **trust, usability, and user adoption**.

The correlation confusion matrix (fig. 4) highlights key relationships between system components and user perceptions. The strongest correlation is between Clarity of hazard indicators and Future Use (0.7), suggesting that the easier it is to spot hazards, the more likely users are to adopt the system. A similarly high correlation exists between UI Usefulness and Future Use (0.65), indicating that a well-designed interface also drives continued use. Trust in the system (0.55) is a factor, but not as strong as clarity or usability. Meanwhile, the low correlation between Clarity and UI Usefulness (0.06) shows that users view the clarity of hazard markers and the overall interface design as separate issues. This shows the importance of trust, clarity, and usefulness in encouraging non-experts to

use such systems regularly.

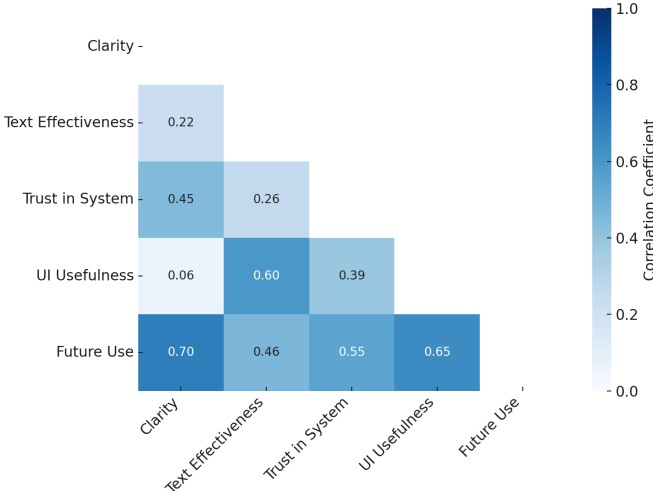

Fig. 4: The confusion matrix shows correlations between the different evaluated components of the system and suggests relatively high relationships across all five metrics (as expected given the system's consistently high scores). For example, the strongest correlation (0.70) is between Future Use and Clarity, suggesting that participants who found the hazards easier to spot were more likely to adopt the system in the future.

### A. Hypothesis Evaluation

In this work, we demonstrated how capabilities of GPT-4o can be utilized in robot-assisted hazard detection, enabling them to perform as a sociable, useful assistant in critical situations. Although the system missed one potential hazard, it successfully identified 18 out of 19 potential hazards in the virtual environment. This supports **Hypothesis H1**: VLMs are able to identify most (more than 80%) of the hazards in the space.

The human feedback gathered from our user study strongly supports our **Hypothesis H2**: people generally feel more at ease navigating a scene with annotations as opposed to one without any. Although we worked with a small sample size, we showed statistically that most users moderately trust and highly accept LLM-supported robots in regulating their safety concerns, suggesting that assistant robots can play a meaningful role in improving safety awareness in high-risk environments like disaster response and industrial settings.

Studies on human-automation trust consistently highlight the need for situational adaptability. This necessitates the active involvement of operators with diverse expertise in the design phase to create more human-centered assistance systems [12]. As such critical scenarios are not easily accessible or repeatable, we emphasize on the positive impacts of our VR-based simulation and hazard analysis in HRI systems.

## VI. VLM REPORTING AND ETHICAL CONSIDERATIONS

The use of large pre-trained models has, or has the potential to have, significant implications regarding privacy, social justice, environmental concerns, and reproducibility. In this section we briefly describe the specifics of the model used and the impact of our work.

**Model specifications:** The system described in this work depends on the use of a Large Vision and Language Model for automatically annotating points of interest in an environment. The VLM was a separate API that was queried to find possible hazards. For the implemented system, we used OpenAI's GPT, specifically model gpt-4o-2024-11-20. No fine-tuning was performed, and no seeds were specified. This model was selected based on its accessibility and performance on a range of queries obtained during pilot studies. We performed all queries during October and November 2024.

**Privacy:** Although we let humans virtually interact with the scene, no human avatar is included in GPT's input images for the Sim2Real process in this study due to privacy and safety concerns. Additionally, human participants are not interacting directly with the VLM, reducing the risk of data leakage based on human inputs.

Future studies and applications will likely incorporate human avatars resembling study participants. Human state is important to the task and could lead to life-threatening consequences if ignored; however, sending images of people to GPT may not be in their best interest in regards to their privacy. Another concern is protecting sensitive information in the environment itself, such as proprietary technology. A potential solution to this is running models locally on the robot; however, this may undermine the performance requirements for accurate hazard annotation, exacerbating accuracy issues. Therefore, more research such as [36] that evaluates how much LLMs are able to keep secrets is essential. Work on smaller multimodal language models like TinyLLaVa [37] may also enable all computation to be done locally, limiting the scope information is shared in.

**Social implications:** Because the VLM is not interacting directly with human participants, concerns such as models that perform unevenly based on voice or skin tone [38], [39] are not relevant to this study, nor is demographic information (or indeed any user specific information) transmitted to the model.

**Hallucinations:** To maximize situational awareness, humans need to rely on the information being presented in the environment. Supporting human trust crucially requires that users be able to observe the system's reasoning in identifying certain items as hazardous, which is currently an opaque operation. To address this concern, in future we will explore using Chain of Thought (CoT) [40] and other XAI [41] techniques. This would be beneficial in cases where similar hazards need to be treated differently in different environments, so as to ensure that the system performs equitably across all environments and scenarios by conducting a series of various experiments in different spaces. Integrating human feedback to fine-tune the VLMs and the pipeline as a whole may help avoid false positives, negatives, and hallucinations in different environments [42]. For this user study, no hallucinations in anomaly detection were presented, as per evaluator and participant analysis; however, in larger scale deployments,

where hallucinations are inevitable, it may become necessary to demonstrate the system's reasoning.

**Environmental:** Environmentally, although detailed figures are difficult to obtain, a conservative estimate is that each of our GPT inferences is responsible for approximately 0.047 kilowatt-hours of electricity [43]. We estimate that the electricity usage of our study was roughly 1.88 kWh (0.047 times the approximately 40 images evaluated). This does not include one-time development costs. Novel CoT models like DeepSeek R1 [44] may help to address environmental concerns by reasoning through what safety standards are relevant for a given scenario with less computational power.

**Omitting the VLM:** Without the use of a vision and language model, this work would have relied on simpler methods of automatically annotating points of interest, possibly with manual involvement (see section II-B). However, because the VLM is core to the contributions of this work, ablation studies were not performed.

## VII. LIMITATIONS AND FUTURE WORK

Integrating AI into any system can introduce uncertainty and skepticism. Additionally, the evolving nature of this field makes it hard to be confident about the system's current state and performance. Some of these limitations are discussed here.

### A. Limitations

VLM-guided robots show promise for handling disastrous scenarios and aiding first-responders; however, improvements can be made in their hazard reporting. For instance, GPT recognized most of the obvious safety issues in the scenario, but did not always consider the interaction between objects, exemplified by its failure to recognize the blockage at the emergency shower (fig. 1a). For this reason, the use of segmentation models like SAM [45] may be warranted to isolate or emphasize certain objects and help VLMs recognize the more intricate interactions in the scenario. One key challenge in hazard detection involves distinguishing between different types of failures.

In hazard detection tasks, two main types of errors can occur: false positives and false negatives. A false positive arises when the system incorrectly classifies a safe object or situation as hazardous, such as mistakenly flagging a properly stored chemical container as a risk. Conversely, a false negative occurs when the system fails to identify an actual hazard, potentially leading to dangerous situations. In high-risk environments, false negatives are particularly concerning, as undetected hazards can have severe consequences. To mitigate this risk, the system should favor over-identification rather than under-identification, ensuring that potential dangers are consistently recognized and addressed.

Although not examined in this study, environmental sounds can play a crucial role in identifying dangerous situations, such as faulty machinery, leaking fluids, or sparking electrical circuits [32]. Future research could incorporate this additional modality by utilizing advanced multimodal models, potentially enhancing the accuracy and detail of annotations. Additionally,

with the rapid emergence of powerful VLMs like LLaVA, Gemini, GPT-4 Vision, and NeVA, the VLM component in our system can be upgraded to improve multimodality, precision, and performance, as well as to reduce hallucinations [46].

The biggest limitation of the work to date is the size of our user study, which was limited to ten participants (with an additional ten participating in a pilot study to inform interface design and prompt development). Although it is our intention to conduct larger studies in future, it is also possible that future large-scale evaluations could utilize platforms such as Prolific to allow more people to experiment with the interface. This is made more difficult by the requirement that participants have access to a VR headset, but as such equipment becomes more widespread, crowdsourcing VR work become a more realistic possibility.

### B. Future Work

There are many avenues for future work that build upon the foundation of the methodology presented in this paper, as well as the strong findings of the user study for utilizing VLMs for effective hazard analysis in the VR interface with POI indicators.

As methodological enhancements, prior work has shown that imposing a coordinate grid on image inputs for VLMs can significantly improve question-answering, image captioning, and segmentation [15]. Future work may improve our system by modifying input images with such a grid or other tools to help the VLM better understand the scene. Since VLMs are starting to support video analysis, future work should also explore whether providing a video to (rather than downsampled still images from the robot's video camera) increases annotation accuracy.

Future work could also explore active hazard response in VR, such as virtually resolving anomalies (e.g., straightening a precariously balanced waste container) to simulate mitigation. Task prioritization and resource management in such hazard response scenarios could also be investigated. Our work to date prioritizes interfaces and situational awareness rather than physical manipulation.

The scope of this study could also be expanded to include team interactions in our hazard identification interface, testing the feasibility of a team of humans interacting with the robot virtually. Furthermore, different personnel require different information in disaster scenarios. For instance, annotations that are useful to firefighters may differ from those needed by paramedics or police officers. Future work could explore generating different classes of annotations tailored to different first responders in team settings.

Disaster scenarios are also dynamic; annotations produced may rapidly become outdated. Future research should explore how autonomous agents can adapt to these changing environments, such as tracking the spread of fire or monitoring the status of injured victims. Modeling these evolving conditions in our simulation framework could allow us to test dynamic analysis informed by hazard message reports.

These results should also be evaluated in real-world conditions through the Sim2Real [47] process before practical use. Reconstructions and digital twins created with Gaussian Splats can accelerate the creation of virtual environments representing hazardous environments. We advocate for the use of such digital twins, as they can represent the status, features, and behavior of their physical twins in real time with high accuracy [48].

## VIII. CONCLUSION

This study supports the cautious integration of large pretrained models in HRI systems [49] as a source of environmental analysis for applicable scenarios. Despite a limited sample size, the results suggest that users found the system satisfactory. Assistant robots can play a meaningful role in enhancing safety awareness in high-risk environments, such as disaster response and industrial settings.

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
