# OpenReview forum: "LLM-Supported Safety Annotation in High-Risk Environments"
_humanrobotinteraction.org/HRI/2025/Workshop/VAM — HRI 2025 Workshop VAM Submission_

### Official Review · Reviewer_JF1g · 2025-02-28

**Rating:** 7
**Confidence:** 5

**Review:**

Paper Review - LLM-Supported Safety Annotation in High-Risk Environments

--Summary--
In this paper, the author(s) present a novel approach for automating detection and annotation of hazardous areas. In this method, images are captured by a robot (or similar mechanism) and then a multi-modal LLM (with vision and text generation capabilities) was used to identify which of the images contained hazards, as well as appropriate information describing the exact hazard and how it could be addressed. The efficacy of the approach was explored through a user study in which 12 participants explored both the unannotated and the annotated environment and then answered some survey items and some open-ended questions about their experience with the system. The results show that the approach was generally effective at identifying and describing the hazards.

Overall, the paper was well-written and made good use of imagery. The work is appropriate for the venue, and would generate good discussion. The work is also quite timely, with a lot of interest and research attention currently being given to the appropriateness of AI. The methods used in this research would likely be of interest to other researchers looking to use VLMs and LLMs to understand and describe spaces.

Feedback
The paper could be clearer in the exact methodology used. I found the abstract and study design sections to be a bit difficult to understand, and I think that clearly describing the entire workflow (or using a diagram) could be very beneficial. E.g.: A robot explores an environment and captures images --> The images are processed using the multi-modal LLM to identify hazard POIs and generate the safety messages --> The JSON data is imported into the VR experience --> The study participants viewed the virtual scenario which contained the annotations in VR --> Participants reflect on the benefit of these annotations.

I would have expected a more rigorous analysis around H1. Currently, the hypothesis states that ‘LLM-supported robots can identify at least 80% of hazardous items in a given space’, but the study only considers a single environment/scenario. It also doesn’t explore the matrix of false positives, false negatives or discuss how the hazards (which were likely used as the ground truth) in this scenario were developed. The hypothesis also currently has an 80% threshold for suitability, but I think that this should be justified. For example, in a dangerous scenario it may be better to tune the system to be overly sensitive instead (e.g., getting more false-positives to prevent false-negatives).

Other
The VLM Reporting and Ethical Considerations section was very interesting, and is something that is often not discussed. I think that the research space would benefit from more papers doing something like this.

---

### Decision · Program_Chairs · 2025-02-26

Accept